# Targeting Apoptotic Pathway of Cancer Cells with Phytochemicals and Plant-Based Nanomaterials

**DOI:** 10.3390/biom13020194

**Published:** 2023-01-18

**Authors:** Atif Khurshid Wani, Nahid Akhtar, Tahir ul Gani Mir, Rattandeep Singh, Prakash Kumar Jha, Shyam Kumar Mallik, Shruti Sinha, Surya Kant Tripathi, Abha Jain, Aprajita Jha, Hari Prasad Devkota, Ajit Prakash

**Affiliations:** 1School of Bioengineering and Biosciences, Lovely Professional University, Phagwara 144411, India; 2Feed the Future Innovation Lab for Collaborative Research on Sustainable Intensification, Kansas State University, Manhattan, KS 66506, USA; 3College of Medical and Allied Sciences, Purbanchal University, Morang 56600, Nepal; 4UNC Blood Research Center, University of North Carolina, Chapel Hill, NC 27599, USA; 5Lineberger Comprehensive Cancer Center, University of North Carolina, Chapel Hill, NC 27599, USA; 6Division of Chemical Biology and Medicinal Chemistry, UNC Eshelman School of Pharmacy, University of North Carolina at Chapel Hill, Chapel Hill, NC 27599, USA; 7School of Biotechnology, Kalinga Institute of Industrial Technology, Bhubaneswar 751024, India; 8Graduate School of Pharmaceutical Sciences, Kumamoto University, Kumamoto 862-0973, Japan; 9Headquarters for Admissions and Education, Kumamoto University, Kurokami, 2-39-1, Chuo-ku, Kumamoto 860-8555, Japan; 10Pharmacy Program, Gandaki University, Pokhara 33700, Nepal; 11Department of Biochemistry and Biophysics, University of North Carolina, Chapel Hill, NC 27599, USA

**Keywords:** cancer, apoptosis, drugs, phytochemicals, inhibitors, nanomaterials

## Abstract

Apoptosis is the elimination of functionally non-essential, neoplastic, and infected cells via the mitochondrial pathway or death receptor pathway. The process of apoptosis is highly regulated through membrane channels and apoptogenic proteins. Apoptosis maintains cellular balance within the human body through cell cycle progression. Loss of apoptosis control prolongs cancer cell survival and allows the accumulation of mutations that can promote angiogenesis, promote cell proliferation, disrupt differentiation, and increase invasiveness during tumor progression. The apoptotic pathway has been extensively studied as a potential drug target in cancer treatment. However, the off-target activities of drugs and negative implications have been a matter of concern over the years. Phytochemicals (PCs) have been studied for their efficacy in various cancer cell lines individually and synergistically. The development of nanoparticles (NPs) through green synthesis has added a new dimension to the advancement of plant-based nanomaterials for effective cancer treatment. This review provides a detailed insight into the fundamental molecular pathways of programmed cell death and highlights the role of PCs along with the existing drugs and plant-based NPs in treating cancer by targeting its programmed cell death (PCD) network.

## 1. Introduction 

Cancer continues to haunt mankind with almost 10 million deaths reported every year [1,2,3]. This multistage disease is caused due to the unchecked cell divisions that lead to the creation of abnormal cells which invade the neighboring tissues. Cancer is generally caused by the interaction of genetic elements with external carcinogens which can be biological (viruses, parasites, and bacteria), physical (ionizing and ultraviolet radiation), and chemical (asbestos, arsenic, and aflatoxin) [4,5]. The complex cancer cell attributes have been organized into different hallmarks such as unlimited replicative potential, metastasis [6], self-sufficiency in growth signal [7], sustained angiogenesis, evading apoptosis, and less sensitivity to anti-growth signals [8]. Even though all the hallmarks have different molecular mechanisms whereby they cause the transformation of a normal cell into a cancerous cell, but they directly or indirectly lead to the process of apoptosis or programmed cell death (PCD). The stability between the proliferation of cells and cell death is crucial for metastasis and invasiveness in cancerous cells [9]. The intrinsic pathway involves internal stress sensing that prefers pro-apoptotic proteins to the anti-apoptotic ones. An extrinsic apoptotic pathway is driven by the cell surface receptor interactions with the ligands [10]. Both pathways connect at the time of caspase activation. The cancer cells surpass PCD via different mechanisms such as anti-apoptotic protein upregulation, downregulation of pro-apoptotic proteins, caspase activity inhibition, etc. Various cancer treatment strategies have been reported around the globe, but they are expensive, and inexpensive treatments are ineffective with various side effects [11]. Thus, the fight against this deadly disease requires a tremendous investment in the research of cancer pathology to develop new, effective, safe, and inexpensive tools or substances as anti-cancer agents. Indigenous societies have long utilized plants for the treatment of diseases [12]. Plants act as useful anti-cancer agents, with more than 3000 plants evaluated for their role in fighting cancer by targeting apoptotic pathways [13,14,15]. In the last couple of decades, plants have been embraced as potent apoptotic inducers in cancer chemotherapy. It is pertinent to mention that 49% of the molecules that have been approved for cancer therapy in the last 60 years belong to natural products [16]. Despite the advantages associated with plants and plant products in cancer treatment, the existing research looks to be inadequate. This prolongs the evaluation of the several effective and safe anti-cancer PCs. PCs also act in synergy with the existing drugs and designed nanomaterials for targeting the apoptotic pathways [17,18]. Several plant-based NPs have been designed through the green synthesis approach for increasing the delivery of drugs to improve the efficiency [19,20]. There have been several articles published that have highlighted the role of PCs as anti-cancer agents [21,22,23]. However, this review article gives a detailed account of the apoptosis and its fundamentals, besides delineating it as a potential drug target in cancer. The aim of this review article is to compile the role of PCs in inducing apoptosis individually, synergistically, and in association with designed NPs, thereby opening new avenues in cancer pathology concerning cancer treatment.

## 2. Apoptosis: Molecular and Cellular Perspective

Apoptosis, programmed cell death, is a series of biochemical processes that are characterized by morphological changes and the death of the cells. The process is pivotal to the cell cycle processes of multicellular organisms that are represented by nuclear fragmentation, cell shrinking, mRNA decay, DNA fragmentation, and blebbing [24]. An adult human and average child (10–12 years old) loses about 50–70 billion and 20–30 billion cells/day, respectively. Apoptosis is an efficiently synchronized cell process that is important throughout the life cycle of an organism. The separation of toes and fingers is one of the common advantages known hitherto that is conferred by PCD at the embryonic stage. Apoptotic bodies are produced during the process of apoptosis which are utilized by the phagocytic cells that prevent their spread which otherwise can be damaging. It is generally accepted that the permeabilization of the outer mitochondrial membrane, a process controlled by the Bcl-2 family of proteins, marks the apoptotic cell’s point of no return. This is characterized by the release from the intermembrane space into the cytosol with key pro-apoptotic functions such as: cytochrome c, apoptosis-inducing factor, endonuclease G, etc. [25]. PCD starts with either an extrinsic pathway (signal from other cells) or an intrinsic pathway (feeble external signals) [26]. Both pathways activate caspase proteins that induce the cell death by degradation of cellular proteins. Even though apoptosis is a biologically important process but the uncontrolled apoptosis leads to atrophy (excessive apoptosis) or cancer (insufficient apoptosis/extreme cell proliferation) [27]. Caspase and Fas receptors are pro-apoptotic, while Bcl-2 family members are anti-apoptotic in action. The apoptotic signaling within a cell initiates as a stress response triggered by a viral infection, radiation, heat, hypoxia, or increase in calcium and fatty acid concentration within the cell [28]. Following an encounter with any of these stimuli, the cell undergoes ordered organelle degradation by the proteolytic activity of the caspases. This leads to the cell’s morphological changes, such as cytoskeleton breakdown, lamellipodia retraction, tight organelle packing, dense cytoplasm, pyknosis, discontinuous nuclear envelope, karyorrhexis, and cell shrinking. Before the disposing of an apoptotic cell, it undergoes a disassembly process which is a three-step process: 1. membrane blebbing, i.e., the formation of irregular membrane buds/blebs regulated by ROCK1 (Rho-associated coiled-coil containing protein kinase 1); 2. membrane protrusion formation; and 3. fragmentation, i.e., the breakdown of a cell into apoptotic bodies. The dead cells are finally removed via efferocytosis by the phagocytic cells as dying cells display a phagocytic molecule known as phosphatidylserine on the surface. The pathways related to cell death are inhibited by the negative regulators of the apoptotic process. The inhibition leads to the development of drug resistance and helps tumors in evading cell death. The ratio of Bax (pro-apoptotic protein) to Bcl-2 (anti-apoptotic protein) is a decisive factor in determining whether a cell dies or lives.

## 3. Apoptosis Activation: Mechanism

The apoptotic nature of the cell is efficiently regulated because it leads to cell death once the process begins [29]. The mitochondrial or intrinsic pathway (IP) and extrinsic pathway (EP) are so far the best understood molecular mechanisms of apoptosis. The IP activation is mediated by the signals generated inside the stressed cells and relies on mitochondria’s intermembrane space (IMS) for the release of proteins [30]. On the other hand, EP is activated by the ligand binding to the cell surface receptors forming a death-inducing signaling complex (DISC). Both IP and EP converge upon caspase-3 and caspase-7 activation. Before the precipitation of the cell death process by enzymes, apoptotic biochemical signals tightly regulate the release of apoptotic proteins. Figure 1 illustrates the intrinsic and extrinsic molecular activation of programmed cell death.

### 3.1. Mitochondria Pathway of Cellular Apoptosis

Mitochondria is important for multicellular organisms because it allows aerobic respiration which is essential for cell survival. This forms the bedrock for the intrinsic apoptotic pathway. Apoptotic proteins target mitochondria that may cause its swelling through the formation of pores in the membrane or may increase mitochondrial membrane permeability allowing the apoptotic effectors to leak [31]. Because of the sensitivity related to IP, tumors arise more through IP than EP. Bcl-2 proteins are the driving forces of IP, and each of its members have homology domains (BH1-BH4) with Bcl-2 that may be one or more [32]. Apoptosis is initiated by the pro-apoptotic proteins such as BH3 having one BH domain [33]. The key effector proteins that are committed to apoptosis include the Bcl-2-interacting mediator (BIM) that is encoded by *Bcl2L11*, p53 upregulated modulator (PUM) of apoptosis encoded by Bcl-2 binding component 3 protein (*BBC3*)BH3-interacting death agonist (BID) that is encoded by *BID,* and other effector molecules that help in the activation of pro-apoptotic proteins or the pore-forming proteins which includes Bcl-2-associated-X-protein (BAX) and Bcl-2-antagonist killer (BAK) [34]. The activation of BAK and BAX at the surface of mitochondria induces an allosteric alteration that enables their oligomerization and results in the formation of macropores in the membrane causing mitochondrial outer membrane permeabilization (MOMP). MOMP ensures the apoptogenic protein release from the IMS. The released proteins activate the caspases in the cytoplasm directly or indirectly. The direct mode of action is studied in the cytochrome c which sticks to the protein scaffold, i.e., apoptotic protease activating factor-1 (APAF1), to transform it into an apoptosome. The indirect mode of action is exemplified in the case of the second-mitochondria-derived activator of caspases (SMAC) that are known to neutralize the caspase-inhibitory proteins [35]. A series of biochemical events lead to the activation of caspase-9, an initiation caspase, which is followed by the activation of caspase-3, -6, and -7, executioner caspases. Although caspase activation is ubiquitous in cell death, it is a principal event for the process of apoptosis because the interception in caspase activation does not rescue dying cells following MOMP under normal physiological conditions. Thus, MOMP is the crucial point where a cell commits irreversibly to undergoing cell death. The cells with restricted MOMP potential are tumorigenic owing to the actions of post-MOMP proteins such as DNases that can mutate the DNA leading to a neoplastic transformation [36]. Besides being the driving force in the mitochondrial protein release, BAK and/or BAX also helps in releasing the mtDNA into the cytoplasm and thereby activating pro-inflammatory signals in the absence of caspases [37]. After MOMP, a process known as met-induced mitochondrial protein (MIMP) is thought to take place, during which components of the mitochondrial matrix are released into the cytoplasm [38]. When Bcl-2 family members tightly regulate MOMP and cause a rapid spillage of mitochondrial IMS proteins, MIMP exhibits unique characteristics in both kinetic and molecular regulation. The slower release of mitochondrial factors, rapid mitochondrial Ca^2+^ uptake, and complete independence from Bcl-2 proteins serve as proof of this. It is interesting that the antiapoptotic Bcl-2 members also prevent cell death by inhibiting IP3-mediated Ca^2+^ transfer from the ER to mitochondria [39]. 

### 3.2. Death Receptor Pathway of Cellular Apoptosis

There are two molecular mechanistic models put forth to understand the extrinsic pathway of apoptosis. These include the TNF-induced (tumor necrosis factor) model and Fas-Fas ligand-mediated model, both of which are associated with extrinsic signals involving TNF-receptor. TNF-α, a cytokine synthesized by macrophages, is a main extrinsic apoptotic mediator [40]. Human cells possess two TNF receptors, namely TNFR-1 and TNFR-2. The binding of TNF-α to TNFR-1 results in the initiation of a biochemical pathway that ultimately causes caspase activation through the membrane proteins [41]. The death domain associated with the TNF receptor known as TNFR1-associated death domain protein (TRADD) and death domain associated with Fas, FAS-associated death domain protein (FADD), can bind to TRAF-2 and thus inhibit the caspase-8 activation. Indirectly, this attachment most likely activates the transcription factors involved in the inflammatory response and cell survival [42]. The signaling cascade via TNFR-1 may also initiate PCD without caspase.

The first apoptosis signal (Fas), also referred to as CD95 or APO-1, a TNF transmembrane protein, binds to the Fas ligand. The association of Fas with FasL leads to the formation of DISC, which is a complex of caspase-8, caspase-10, and FADD. In type I cells (branched cells with multiple cytoplasmic plates), the caspase-8 which is preprocessed directly activates other caspase family members which initiates the apoptosis execution [43]. On the other hand, in type II cells (progenitors of type I), Fas-DISC triggers a feedback system that coils into the release of increased pro-apoptotic factors from mitochondria, thereby increasing the degree of caspase-8 activation [44].

## 4. Hallmarks of Apoptosis

### 4.1. Morphological Indicators

The change in cellular morphology during PCD is one of the defining features that characterizes the process of apoptosis. The microscopic examination in the initial stages of apoptosis is symbolized by nuclear material condensation and margination of chromatin along the nuclear envelope [45]. This is followed by nuclear fragmentation and intricate folding of the nuclear membrane. The early stage apoptosis on the surface of cells is characterized by apoptotic cell detachment from neighboring cells, formation of cytoplasmic blebs, and loss of microvilli. As apoptosis advances, the cellular cytoplasm undergoes a peculiar condensation that leads to the compaction of cell organelles and then forms the apoptotic bodies which are membrane-bound. Finally, these are consumed by macrophages or nearby parenchymal cells.

### 4.2. Biochemical Indicators

The evaluation of biochemical indicators of PCD have certain limitations given the fact that cell death in tissues is asynchronous in nature; therefore, only a fraction of the cells undergoes PCD at a particular point in time. In addition, the cells are instantly phagocytosed once they enter the apoptotic process [46]. To overcome the hindrances that make biochemical characterization difficult, the model system has been expanded, which depends on the hormonal enrichment of cells. One common model system is the immature thymocyte in which glucocorticoid hormones act as apoptotic activators. One of the biochemical events studied by using the thymocyte model system is the genome degradation process in apoptosis [47]. Conradt et al. studied the degradation of DNA through programmed cell death, which occurs via the cleavage of genomes at the linker region in *Caenorhabditis elegans* [48]. Some of the studies demonstrated that PCD is mediated through the glucocorticoid receptor pathway (GRP) (Table 1). The studies were based on the fact that endogenous and exogenous glucocorticoids activate endonuclease activity, the antagonist nature of glucocorticoid can block the apoptotic response partially, a functional GR is required for the apoptosis activation, and non-glucocorticoid hormones were unable to evoke endonuclease activity.

## 5. Significance of Apoptosis

Apoptosis acts in association with the process of mitogenesis and cellular differentiation to organize cellular function which ultimately orchestrates physiological functioning. Initially, PCD plays a significant role at the time of intrauterine development. Figure 2 illustrates the significance of the normal and abnormal apoptotic process. It shapes the sculpture and organizes the interdigital webs of toes and fingers [49]. Moreover, apoptosis is a fundamental process for the usual overturn of intestinal cells, regression of tail in tadpoles via thyroxine dependency, insect cell metamorphosis, eye lens development in chicks, prostate epithelium regression, mammary gland, thymus gland, and adrenal gland. Thus, the role of apoptosis is quite ubiquitous in human biology. PCD is also pivotal as a fetal abnormality determinant. One of the studies reported that the wild-type p53 embryo of mouse aborts promptly following the induction of teratogenesis with radiation, but p53-null embryos do not. The biological systems such as the nervous system and immune system develop in response to the cellular overproduction and then followed by PCD of the cells that are unable to develop productive synaptic connections or functional antigen specificities [50]. In a normal adult human being, about 10 billion cells undergo apoptosis to maintain the cellular balance within the body. Thus, regular homeostasis is not only a passive process but highly regulated through PCD. However, with aging, the apoptotic reaction to DNA damages lessens and is weakly controlled, thereby paving the way for degenerative diseases. The induced sensitivity in apoptotic responses also contributes to cancer susceptibility.

### 5.1. Apoptosis: A Disease Paragon

Apoptosis combats infections by helping with the removal of pathogenic niches inside the cell and manages cancer prevention by eliminating the nascent neoplastic cells in developed organisms. However, aberrant PCD has been reported to add to the pathologies associated with long-standing degenerative diseases and acute injuries [27]. Both intrinsic and extrinsic apoptotic pathways play a substantial role in immune response activation and thereby help in the fight against pathogens and cancer. The pathogens specific to the intracellular environment when released by the dying cells are consumed by the neutrophils and macrophages that are in the vicinity, which subsequently leads to chemokine and cytokine secretions. Dying cells release antigens, pathogen-associated molecular patterns (PAMPs), and damage-associated molecular patterns (DAMPs), which are identified and consumed by the specific dendritic cells and allows antigen-presenting cells (APCs) to prime T-lymphocytes, which enables them to recognize and damage the other infected cells, besides helping B-cell differentiation into antibody-producing plasma cells [51].

### 5.2. Apoptosis: A Cancer Cell Hallmark

Finding new anticancer treatments that target the apoptotic pathway is an intriguing strategy. Numerous mutations are found in intrinsic and extrinsic pathways in cancer, which enable the cells to avoid apoptosis. An all-encompassing cancer treatment would be possible if an apoptotic pathway could be targeted and activated [52]. Apoptosis is best characterized and studied in synchrony with cancer. Bcl-2, an anti-apoptotic protein family, was discovered after its overexpression and translocation in B cell follicular lymphomas. The copy number (CN) amplifications acquired somatically of a specific gene marker (loci) encoding Bcl-XL and MCL-1 (pro-survival proteins) are found in 3% and 10% of human cancers [53]. The unavailability of BIM (Bcl-2-interacting mediator), a pro-apoptotic protein, and the availability of overexpressed Bcl-2 were reported to increase *c-myc* oncogene-mediated lymphoma development which strongly indicates that apoptosis evasion promotes cancer. It is pertinent to mention that defective PCD does not alone aid in developing tumorigenesis, as only 5% of mice overexpressing Bcl-2 over eighteen months developed lymphoma, rather only the PCD inhibition aids in tumor development, which allows cells to live that would normally die. The reports suggest that the anti-cancer agents induce the IP for apoptosis, whereas direct activation of EP may not be needed to kill the cells downstream of anti-neoplastic agents [52]. However, the upregulation of TRAIL and FAS receptors mediated by p53, a tumor suppressor, may stimulate the malignant cell to the death receptor ligands. The ligands are efficiently expressed by cytotoxic lymphocytes that are activated and as a result, the ensuing DR-induced killing contributes to the in vivo cancer therapy. The studies on cell lines and mice for gene knockouts have shown the induction of apoptosis by anti-cancer agents in a BH-3-protein-dependent manner [54]. The PCD induction by dexamethasone, a glucocorticoid, depends on BIM. BIM is known to be suitable for tumor cell killing via oncogenic kinase inhibitors such as EGRF mutant, BCR-ABL, or B-RAF mutant. In HeLa cells, apoptosis is stopped by specific cellular proteins that are inhibitory in action and act by targeting retinoblastoma-associated tumor-suppressing proteins [55]. These proteins are known to regulate the cell cycle under normal conditions, but when attached to a protein of an inhibitory nature, these are rendered inactive. Human papillomavirus (HPV), forming a cervical tumor, expresses HPV-E6 and HPV-E7 inhibitory proteins. HPV-E6 inactivates p53 which regulates the cell cycle and acts as a tumor suppressor. On the other hand, HPV-E7 attaches to the tumor-suppressing proteins of retinoblastoma and reduces the cell division control ability. These two proteins with inhibitory action are somewhat responsible for the immortality of HeLa cells. As a result of advancements in cellular and molecular biology, immunology, and genomics leading to the discovery of numerous novel oncogenes, tumor suppressor genes, and immunologic and therapeutic targets, significant progress has been made with more new therapeutic drugs, including chemotherapy, targeted therapy, and immunotherapy, being approved for cancer therapy. The effectiveness and clinical applications of therapeutic drugs are nevertheless constrained by drug-induced toxicities and drug resistance. To increase efficacy and lessen toxicity, novel anticancer agents must be discovered and developed immediately.

### 5.3. Apoptosis: Role in Degenerative Diseases

The defects in controlled cell death may also lead to excess apoptosis which results in degenerative disorders, tissue damage, and hematological diseases. It is noteworthy that neurons depending on mitochondrial respiration go through PCD in diseases such as Parkinson’s and Alzheimer’s [56]. Interestingly, scientists have established that there exists an inverse epidemiological comorbidity relationship between cancer and neurodegenerative diseases. HIV progression is precisely an outcome of uncontrolled and excess apoptosis. This is because of the defects in the molecular signaling pathways that control the activity of Bcl-2 proteins [57]. The overexpression of BIM apoptotic proteins or their reduced proteolysis results in cell death, which ultimately results in a series of pathologies owing to the BIM activity site. Oxygen supply disruption, nutrient provision loss, and cellular waste removal inability lead to tissue injury and promote cell death during blood vessel blockage through embolism or thrombosis in an ischemic event.

## 6. Phytochemicals (PCs) as Strong PCD Inducers

Secondary metabolites of plants are considered to be affordable, safe, and efficient natural compounds exhibiting diverse bioactivities including anti-cancer properties. The secondary metabolites are generally produced by a plant as organic molecules without being essential for plant growth, reproduction, and development. These are categorized according to the pathways that are involved in their synthesis. These include terpenoids, phenolics, and alkaloids. The well-known alkaloids as anti-cancer agents include vincristine, camptothecin, and vinblastine; phenolics include resveratrol and curcumin; and terpenoids include gamma-tocopherol and lycopene [58]. These compounds act synergistically or independently while targeting any cancer cell hallmark such as inhibiting cell cycle proteins involved in cancer progression, microtubule assembly, angiogenesis, or inducing apoptosis [59]. PCs do not allow cancer cells to suppress pro-apoptotic proteins that help cancer cells to survive in nutrient and environmental stress. However, the apoptotic evasion of a cancerous cell does not give rise to PCD-resistant cells; thus, they display higher sensitivity to ionizing radiations and cytotoxic chemotherapy [60]. Some of the major classes of PCs have been delineated below with their potential role in cancer therapy by targeting apoptotic pathways.

### 6.1. Vinca Alkaloids

These are best known for their anti-microtubule and anti-mitotic role and are obtained mostly from the plant of genus *Vinca. Catharanthus roseus* (syn. *Vinca rosea*), commonly known as Madagascan periwinkle, which is the source of PCs such as vindoline and catharanthine. These ultimately help in producing vinca alkaloids such as vincristine, vinblastine, and leurosine. Vinorelbine, a semi-synthetic therapeutic agent, is also prepared by using either catharantine or vindoline. They are cytotoxic in nature, and they act on the cell cycle, thereby inhibiting the potential of rapid division in cancer cells. They prevent microtubule formation, which is pivotal in cellular division, by affecting the tubulin monomers. Mavrogiannis et al. researched the vinca alkaloids to study their effect on the tumor-suppressive and oncogenic microRNAs that target the apoptosis in breast cancer cell lines (BT-20). On treating BT-20 cells with vinblastine, vinrelbine, and/or vincristine, there was a significant upregulation of p53. However, there was no change in the expression levels of the Bcl-2 family [61]. Another study found that vinca alkaloid-induced apoptosis is inhibited by glucocorticoids in breast cancer cells (BCap37) and epidermoid tumor (KB) cells without G_2_M arrest, indicating that apoptosis induced by vinca alkaloids may occur without pathway dependence to halt cellular progression [62]. Groninger et al. demonstrated the in vivo activation of caspase-3 and -9 in leukemic cells after giving vincristine doses [63]. Shinwari et al. also studied the role of lomustine and vincristine in inducing apoptosis and p21 upregulation in blastoma cells. They found both the compounds trigger apoptosis via the mitochondrial pathway, which results in the decrease in Bcl-xl and Bcl-2 anti-apoptotic proteins [64].

### 6.2. Epipodophyllotoxins

Epipodophyllotoxins are naturally occurring anti-cancer plant compounds. The roots of *Podophyllum peltatum* (Mayapple) are the main source of epipodophyllotoxin derivatives including teniposide and etoposide. Etoposide is administered in cancers such as Ewing’s sarcoma, testicular cancer, lung cancer, lymphoma, and Kaposi’s sarcoma. It is known to form a complex with DNA and topoisomerase II, which stops the DNA strands from re-ligation and thereby causes a break in the DNA strands [65]. Since cancer cells divide more rapidly, they depend more on topoisomerase II than normal cells. This causes problems in the synthesis of DNA and encourages the PCD of the cancer cells [66]. On the other hand, teniposide is useful in treating Hodgkin’s lymphoma, brain tumors lymphocytic leukemia, and other cancers. This acts by causing breaks in the DNA by inhibiting the topoisomerase II which ultimately leads to the IP or EP of PCD.

### 6.3. Taxanes

Taxanes, a terpene class, are found in plants of the genus *Taxus.* Cabazitaxel, an FDA-approved anti-cancer agent, paclitaxel, and docetaxel are some of the members of the taxane family that are used as chemotherapeutic agents. They are mostly obtained from natural sources but many are semi-synthesized [67]. Since taxanes are insoluble or poorly soluble in water, their formulations are often challenging. As an anti-cancerous agent, they target the activity of microtubules. They are known to inhibit the cell division process by preventing de-polymerization in microtubules, thus acting as mitotic inhibitors [68]. In comparison to the taxanes, the above-discussed vinca alkaloids act by preventing spindle formation by inhibiting the polymerization of tubulins. Thus, both are designated spindle poisons and help in inducing apoptosis in rapidly dividing cancerous cells.

### 6.4. Camptothecin

Camptothecin, a natural anti-cancer agent, was first isolated from the stem and bark of *Camptotheca acuminata* (happy tree). Camptothecin has shown significant anti-cancer properties in trials against gastrointestinal tumors, colon cancer, breast cancer, and lung cancers [69]. Based on this, scientists have formulated numerous camptothecin analogs that are approved for the use of cancer therapies including irinotecan, belotecan, trastuzumab, and topotecan [70]. It acts by binding to the DNA and topoisomerase II forming a ternary complex. This causes damage to the DNA because it stops the re-ligation process and ultimately initiates apoptotic pathways in the cancerous cells. The binding of camptothecin with DNA and the topoisomerase enzyme forms a hydrogen bond and it shows selective toxicity towards some cells. Table 1 highlights the significance of numerous PCs as anti-cancer agents acting directly or indirectly through PCD signaling pathways.

## 7. Combinatorial Use of PCs and Common Drugs as Pro-Apoptotic Agents

Cancer treatment with PCs is not effective because it is not possible to target cytotoxic effects in cancerous cells. Therefore, a combinatorial strategy of these molecules with conventional chemotherapeutics helps to increase toxicity in cancer cells and reduce therapeutic doses and toxicity levels. In this regard, several studies have highlighted the synergistic usage of PCs with a potential role in targeting the apoptotic pathways of cancer cells [101]. Resveratol has been successfully utilized as an adjuvant of various drugs, such as temozolomide, paclitaxel, and doxorubicin in mice models. Curcumin, a chemosensitizing agent, when combined with docetaxel showed a substantial decrease in drug resistance in breast cancer cells [102]. Similar kinds of results were reported when curcumin was combined with cisplatin, irinotecan, and vincristine, thereby improving the efficacy of these drugs. Gingerol in combination with cisplatin and doxorubicin also enhances the sensitivity of gastric and liver cancer cells, respectively. Other compounds that have been used in combination with conventional drugs are listed in Table 2.

## 8. Hinderances in Using PCs as Pro-Apoptotic Agents in Cancerous Cells

The development of PCs as pro-apoptotic agents requires strong evidence of quality, safety, and efficacy during the clinical trials. Although the listed compounds have shown notable effects on the induction of apoptosis, some limitations need redressal before advancing to the clinical stage. The poor solubility, absorption, side effects, and poor penetration are among the major concerns in targeting cancer cells by PCs. For example, camptothecin, podophyllotoxin, and colchicine-like compounds have been the least used due to the side effects associated with them [113]. Besides these limitations, the discovery and development of PCs employed clinically also face challenges related to extraction, optimization, and characterization. Thus, novel analytical tools and computational methodologies are needed to facilitate identification, and optimization of anti-cancer molecules. Some PCs may also be present in lesser quantity. To overcome this hinderance shortcoming related to absorption, metabolism, and stability, lipid-based formulations (liposomes, microencapsulation, and nano-emulsion) are considered suitable methods to improve solubility, target specificity, stability besides reducing the quantity of PCs in order to achieve the efficient therapeutic levels [114]. Vinca alkaloid administration has been enhanced by using micelles or liposomes (Patent number: US4952408A & US8765181B2). The entrapment of active molecules allows controlled release and prevents the bioactive molecules from external factors that could interfere in the performance [115]. Among them, Marqibo, an FDA-approved liposome for adult leukemia, is a common example. Several efforts have been made in this field but only a few formulations have reached the stage of clinical trials. This includes the improvement of paclitaxel solubility. Hydroxypropyl-methacrylamide copolymer-PTX, developed by Pfizer, was stopped in phase I owing to its higher toxicity in rats. It has been reported that resveratrol has a circulation half-life of several minutes while quercetin is present in low concentrations making it an insufficient anti-cancer agent [116]. Proliposome-formulated gingerol has shown higher in vitro inhibition of cancer cells and high in vivo oral bioavailability [117]. Of late, hyaluronic-acid-conjugated mesoporous silica NPs were employed in the drug delivery of curcumin to enhance the anti-cancer activity. The results significantly showed that the delivery system improved anti-cancer activity both in vitro and in vivo [20]. Several strategies are being performed to enhance the activity and availability of chemotherapeutic drugs, but there is still a need to upscale the clinical trials for most of these compounds in the future.

## 9. Plant-Based Nanomaterials Targeting Apoptosis and Other Cancer-Associated Pathways

NPs with typical sizes less than 100 nm have been extensively used in a variety of industrial applications, including electronics [118], optics [119], textiles [120], pharmaceuticals [121,122], environmental remediation [123,124], and wastewater treatment [125,126]. The effective use of NPs in biomedical disciplines such as bioimaging [127], theragnostic [128], tissue engineering [129], biomolecular detection and diagnostics [130], and drug delivery [3] has been described by several studies. Plant extracts or tissues serve as an essential and effective component for the green synthesis of NPs [131]. Phytochemicals such as alkaloids, proteins, sugars, flavonoids, terpenoids, polysaccharides, and enzymes, besides acting as capping and stabilizing agents, may also function as reducing agents for the synthesis of NPs [132,133]. The plant-based green synthesis of NPs approach relies on the interaction of PCs as a capping agent (stabilizing or reducing agents) with specific metal or metal oxide for their synthesis [134]. The general scheme for the synthesis of plant-based NPs is described in Figure 3. These bioengineered NPs have several advantages that make them strong candidates for anticancer drugs or drug delivery [135]. Gold, copper, silver, iron, and titanium are among the metals that are commonly used in green synthesis NPs. Such plant-based metal NPs have gained significant interest in the scientific community because of their nontoxic, eco-friendly, cost-effective, and enhanced optical properties (Table 3) [136]. Additionally, plant-based NPs are more biocompatible than those synthesized chemically. Figure 3 is an illustration of NP synthesis and its potential role in cancer therapy.

### 9.1. Plant-Based Silver NPs

Silver nanoparticles (AgNPs) are considered ideal candidates for cancer therapy owing to their small size and potential to cause cell death by rupturing double-stranded DNA, oxidative stress, and chromosomal instability [19]. Although smaller AgNPs (10 nm) are more likely to cause cellular toxicity as they can penetrate cells easily, larger AgNPs (100 nm) are more efficient at producing these effects. In mammalian cells, AgNPs have been found to cause cell toxicity via a number of mechanisms, including the production of reactive oxygen species (ROS), damaging the cell membrane, and inducing DNA replication by taking up free Ag ions [137,138]. Green synthesized AgNPs have been found to induce cancers such as liver cancer, breast cancer, Ehrlich ascites carcinoma, colon adenocarcinoma, and lung cancer. AgNPs synthesized using crude extract of *Syzygium aromaticum* have been found to be effective against A549 lung and MCF-7 breast cancer cell lines by disrupting the plasma membrane and inhibiting cell growth [139]. Similarly, green synthesis of AgNPs using *Nepeta deflersiana* has revealed its efficiency against human cervical cancer cells by inducing cell cycle arrest and oxidative stress followed by the death of tumor cells [140]. In another study, AgNPs synthesized from cotton leaf induced the mitochondrial-dependent death of lung adenocarcinoma cells [141]. The anticancer and apoptotic potential of AgNPs synthesized from *Fagonia indica* against the inhibition of the growth of MCF-7 cells has been found to be concentration dependent (IC_50_ 12.35 μg/mL).

The results revealed the production of ROS that causes oxidative stress in MCF-7 cell lines. Moreover, these bioengineered NPs induce nuclear condensation and cell membrane damage, followed by cell death [142]. In another in vitro study, the cytotoxic effect of AgNPs synthesized from root extracts of *Beta vulgaris* on normal (CHANG) and cancerous human hepatic cells (HuH-7) has been investigated [143]. The results showed that AgNPs induce higher ROS in HuH-7 cells than in normal cells. Moreover, the higher concentration of AgNPs causes the breakage of DNA strands and increased apoptotic cell count. Biosynthesis of AgNPs using the leaf extract of *Ginkgo biloba* has been evaluated for anticancer activity against cervical cancer (CCa) [144]. It was found that green synthesized AgNPs inhibit cell proliferation and induce apoptosis by increasing intracellular ROS levels. The results also revealed that treatment of AgNPs on CCa cells induces the activation of the caspase-dependent mitochondrial apoptotic pathway. AgNPs synthesized from Alginate extract obtained from *Sargassum vulgare* have also been evaluated for their selective toxicity against cancer cells (HL60 and HeLa cell lines) [145]. The synthesis of AgNPs from tamarind fruit shells has been found to induce apoptosis in human breast cancer cells via DNA impairment and damage. The AgNPs show dose-dependent anticancer activity and inhibition of ROS production generated through the mitochondrial electron transport chain [146].

### 9.2. Plant-Based Gold NPs

For the last couple of decades, gold nanoparticles (AuNPs) have been extensively studied as photothermal agents, drug delivery agents, and radiosensitizers for cancer therapy. The characteristic properties of AuNPs, such as their ability to bind amine and thiol groups and surface plasmon resonance (SPR), make them an ideal candidate for biomedical applications [147]. Several studies are now being conducted on nanoparticle functionalization, and it is moving quickly toward creating biocompatible, multifunctional particles for application in the treatment and diagnosis of cancer [148,149]. GNPs offer a variety of advantages that make them desirable for use in cancer treatment. Due to their small size, they can penetrate cells, and their enhanced permeability and retention effect (EPR effect) enables them to accumulate at tumor sites [150,151]. Additionally, due to their high atomic number, GNPs absorb more kilovoltage X-rays and provide more contrast than conventional agents. When subjected to light with a particular energy, they resonate and produce heat that can be exploited for the photothermal treatment of tumor cells [152].

The synthesis of AuPs using plant extracts has been the center of interest for many researchers because of their wide range of biomedical applications such as antioxidant, antidiabetic, antimicrobial, and anticancer properties. Plant extracts have been found to reduce Au ions to AuPs effectively. Green synthesis of AuNPs using the leaf extract of *Zataria multiflora* has been reported to show anticancer activity against human cervical cancer cells (HeLa cells) [153]. It was found that AuPs can inhibit cell proliferation and induce apoptosis in a dose-dependent manner by activating caspase-3 and -9 in Hela cells. The synthesis of AuNPs using the rhizome extract of *Curcuma wenyujin* has been assessed for its apoptotic effect against human renal cell carcinoma (A498 cell line) [154]. The results indicated that the synthesized AuNPs were significantly sensitive to A498 cells with CC50 values of 25 µg/mL. Moreover, the results showed the generation of ROS, nuclear damage, mitochondrial membrane damage, and finally, apoptosis in A498 real cancer cells. In another study, AuNPs were synthesized using the leaf extract of *Rabdosia rubescens* and were investigated for their anticancer properties against lung carcinoma A549 cell lines [155]. It was found that IC50 doses of AuPs at 25 µg/mL and 50 µg/mL induce apoptosis, thereby confirming biosynthesized AuNPs as a potent agent for the treatment of lung cancer. Anticancer properties of AuNPs synthesized from *Marsdenia tenacissiam* have been found to inhibit cell proliferation in the A549 cell line, thus acting as an ideal candidate for lung cancer therapy [156]. The green synthesized AuNPs induced dose-dependent toxicity against A549 cells, thereby inhibiting cell growth. Moreover, the treatment of AuNPs in A549 cells activated caspase expression and downregulated the anti-apoptotic protein expression. The ethanol extract of clove *(Syzygium aromaticum)* has been used for the green synthesis of AuNPs and evaluated for their anticancer potentiality against the SUDHL-4 cell line [157]. AuNPs reduced the growth and viability of the SU-DHL-4 cell line, and an increase in apoptosis was observed. ROS generation in mitochondria was moderately and significantly increased (time- and dose-dependent) upon treatment of green synthesized AuNPs with SU-DHL-4 cells. The abrogation of apoptosis by the antioxidant *N*-acetyl-l-cysteine (NAC) and the augmentation of apoptosis by the GSH-depleting agent buthionine sulfoximine (BSO) showed that ROS played a significant role in the apoptosis of lymphoma cell lines SUDHL-4.

Cervical cancer is one of the most common cancers, with high mortality rates among women. The aqueous leaf extract of *Alternanthera sessilis* has been used as a reducing agent for the green synthesis of AuNPs, and its anticancer activity against cervical cancer cells (Hela) has been investigated [158]. It was found that AuNPs at 1–15 µg/mL caused a cytotoxic effect on cancer cells. The results also revealed pronounced apoptosis at 10–15 µg/mL concentration of AuNPs. The biosynthesis of AuNPs using ethanolic extract of *Siberian ginseng* has been found to induce apoptosis in melanoma cells (B16) by decreasing the mitochondrial membrane potential (MPP) and enhancing the generation of ROS in mitochondria [159]. Moreover, a significant increase in apoptotic Bad, Bid, Casp-9, and -3 genes and a decrease in antiapoptotic Bcl2 genes were observed in cells treated with green synthesized AuNPs.

### 9.3. Plant-Based Zinc Oxide NPs

Zinc oxide nanoparticles (ZnO NPs) have been effectively used as drug carriers to load and deliver drugs to their intended sites [160]. Additionally, the potential anticancer and antibacterial properties make them appealing to metal oxide NPs in the biomedical field. The ability of ZnO NPs to produce reactive oxygen species (ROS) and cause apoptosis makes them one of the ideal candidates for cancer treatment [161]. It is well recognized that all bodily tissues, including the brain, muscle, bone, and skin, contain significant amounts of zinc, an essential trace element. Zinc participates in the metabolism processes and is a crucial component of several enzyme systems essential for synthesizing proteins and nucleic acids, hematopoiesis, and neurogenesis [162,163]. Nano-sized ZnO particles make zinc easier for the body to absorb. ZnO NPs, which are substantially less toxic and more cost-effective than other metal oxide NPs, have suitable medicinal applications in the treatment of cancer [164], diabetes [165], drug delivery [166], and other bacterial and inflammatory conditions [167,168,169].

Various studies have been conducted to investigate the apoptotic properties of NPs synthesized from plant-based materials. Green synthesis of ZnO NPs using the leaf extract of *Eclipta prostrata* have been found to be significantly effective against human liver carcinoma cells (Hep-G2) [170]. The ZnO NPs of *Eclipta prostrata* have been found to cause a cytopathic effect on the Hep-G2 cell line in a dose-dependent manner. The results confirmed the cytotoxic and apoptotic effect of ZnO NPs at a 100 mg/mL concentration. ZnO NPs synthesized from Rehmanniae Radix have been investigated for their anticancer properties against osteosarcoma cell lines (MG-63) [171]. It was found that green synthesized ZnO NPs inhibited MG-63 cell growth at high doses, increased ROS generation, decreased MMP, and ultimately enhanced apoptotic proteins (Bax, caspase-3, and caspase-9) levels. Thus, ZnO NPs could be used as a potential candidate for inducing apoptosis in bone cancer cells. Green synthesis of ZnO NPs using fruit extract of *Borassus flabellifer* is one of the potential candidates for inducing dose-dependent cytotoxicity and apoptosis on the breast (MCF-7) and colon (HT-29) carcinoma with IC_50_ 0.125 μg/mL [172]. The biosynthesized ZnO NPs from *Marsdenia tenacissima* has been assessed for their anti-cancer activity against laryngeal cancer Hep-2 cell line [173]. The results confirmed the cytotoxic effect and apoptotic potential of ZnO NPs on Hep-2 cells. The treatment of ZnO NPs on Hep-2 cells resulted in the enhanced generation of ROS, nuclear damage, and disruption of MMP. Apoptosis potential was confirmed by decreased antiapoptotic protein Bcl-2 and increased proapoptotic proteins such as Bax, caspase-9, and caspase-3. In another study, ZnO NPs synthesized from bark extract of *Cinnamomum verum* were found to be effective in inhibiting oral cancer KB cells. The green synthesized ZnO NPs at a concentration of 15 and 20 μg enhanced ROS generation and diminished MMP. Moreover, enhanced caspase activity and decreased cell adhesion triggered apoptosis in KB cells when treated with ZnO NPs synthesized from *Cinnamomum verum*, thereby making them an ideal candidate for the treatment of oral cancer [174]. The leaf extract of *Solanum nigrum* has been used for the synthesis of ZnO NPs and investigated for their anticancer properties against cervical cancer (HeLa cell lines) [175]. The results revealed that ZnO NPs of *Solanum nigrum* exhibit anticancer properties against cervical cancer through the apoptotic pathway. Moreover, the results indicated a dose-dependent cytotoxic effect on cancer cells by inhibiting β-catenin and increasing caspase-3, caspase-9, and p53 levels. ZnO NPs synthesized from the leaf extract of *Morus nigra* have been found to be effective against adenocarcinoma gastric cell line (AGS) [176]. The treatment with ZnO NPs synthesized from *Morus nigra* induced cell death by enhancing ROS formation and cytotoxicity and decreasing MMP and cell viability. Moreover, the green synthesized ZnO NPs of *Morus nigra* induced apoptosis in AGS cell lines by upregulating proapoptotic proteins, downregulating anti-apoptotic proteins, and arresting the cell cycle.

### 9.4. Plant-Based Copper NPs

In contrast to other metal NPs, copper nanoparticles (CuNPs) can be synthesized with high production yields using quick and straightforward experimental techniques and economic reaction conditions [177,178]. Their versatility as therapeutic, pharmacological, and medicinal agents is aided by the wide variety of nano dimensions and surface-to-volume ratios that they possess [179,180]. Additionally, CuNPs may be conjugated with various biomolecules, such as enzymes and proteins, to change their surface characteristics, making them effective DNA-cleaving agents and anticancer treatments [181]. However, the synthesis of CuNPs for biomedical applications has been a challenging task since it is difficult to maintain their stability at an ambient temperature. The main reason behind this is that copper under normal conditions gets readily oxidized [182,183]. Despite this, the synthesis of CuNPs from plant material has been explored for their anticancer properties.

The synthesis of CuNPs from floral extract *Quisqualis indica* has been found to induce cytotoxicity and apoptosis in B16F10 melanoma cells [184]. It was found that CuNPs of *Quisqualis indica* induce cytotoxicity via ROS generation, release lactate dehydrogenase (LDH), and downregulate intracellular reduced glutathione (GSH) content. Similarly, CuNPs synthesized from *Phaseolus vulgaris* induce apoptosis and the growth inhibition of cervical cancer cells [185]. The results showed dose-dependent intracellular ROS generation and decreased cervical carcinoma cells. In another study, CuNPs synthesized from the leaf extract of *Olea europaea* as a reducing agent have been found to be effective in stimulating dose-dependent cell growth arrest and cell death via activation of the apoptotic pathway in breast cancer (AMJ-13) and ovarian cancer (SKOV-3) cells [186]. The synthesis of CuNPs using the leaf extract of *Ficus religiosa* was also found to induce a cytotoxic effect against lung A549 cancer cells [187]. Mitochondrial depolarization and upregulation of ROS were found to induce apoptosis in cancer cells. The green synthesis of copper oxide nanoparticles (CuONPs) from the leaf extract of *Azadirachta indica* has been investigated for anticancer properties against MCF-7 and Hela cells [188]. The treatment of green synthesized CuONPs induced toxicity, enhanced ROS generation, and increased DNA fragmentation observed in cancer cells. Moreover, an increase in the level of apoptotic protein markers such as Bax, caspase-9, caspase-8, caspase-3, P^38^, and cytochrome was observed in cancer cells. In another study, the biosynthesis of CuNPs from the leaf extract of *Ziziphus zizyphus* was examined for its effect on human renal carcinoma A498 cells [189]. The results indicated the potent toxicity of CuNPs and enhanced intracellular ROS generation followed by apoptosis in A498 cells. Apoptosis was confirmed by the upregulation of *Bid*, *Bax*, and caspases (3 and 9) and the downregulation of Bcl-2 gene expression in A498 cells treated with CuNPs of *Ziziphus ziziphus.* Moreover, inhibition of the mTOR and PI3K/Akt pathway also confirmed apoptosis in CuNP-treated A498 cells.

### 9.5. Plant-Based Platinum and Titanium dioxide NPs

Several studies have been conducted to synthesize platinum nanoparticles (PtNPs) via green synthesis, highlighting the importance of plant extracts in reducing platinum sources to NPs and their applications in various scientific fields, including cancer therapy. PtNPs of the peel of *Punica granatum* (pomegranate) have been found to inhibit the proliferation of the MCF-7 human breast cancer cell line [190]. Treatment with PtNPs of *Punica granatum* induced apoptosis via the molecular DNA fragmentation of cancer cells. The synthesis of PtNPs using *Tragia involucrata* has been found to induce mitochondria-associated apoptosis in HeLa cells via the enhanced production of ROS [191]. In another study, the synthesis of *Azadirachta indica*-mediated PtNPs has been found to induce time- and dose-dependent apoptosis on HEK293 cells [192]. The treatment of green synthesized PtNPs of *Azadirachta indica* on HEK293 induced toxicity by increasing the level of caspase-9, depolarizing MMP and DNA fragmentation. Green synthesis of PtNPs and palladium nanoparticles (PdNPs) using a tuber extract of *Gloriosa superba* has been investigated for their anticancer properties against human breast adenocarcinoma (MCF-7 cell lines) [193]. The treatment with PtNPs and PdNPs (200 µg/mL) of *Gloriosa superba* for a period of 24 h induced apoptosis on MCF-7 cancer cells up to 12.32% and 31.3%, respectively.

Titanium dioxide nanoparticles (TiO_2_ NPs) have been considered to be a promising candidate for photodynamic therapy (PDT), which is used to treat a wide range of diseases by producing ROS, which causes cell death. After UV radiation exposure in aqueous conditions, the inorganic compound TiO_2_ can release ROS [194]. TiO2 NPs and their hybrid biomolecules, as photosensitizing drugs for treating cancer and bacterial infections, have been the center of interest for many researchers across the globe [195]. However, limited studies have been conducted on green synthesized TiO_2_ NPs for their anticancer properties. TiO_2_ NPs synthesized from the leaf extract of *Zanthoxylum armatum* as a reducing agent have been found to be effective against 4T1 breast cancer cells [196]. The synthesized TiO_2_ NPs induced dose-dependent apoptosis in cancer cells through ROS generation. In another study, TiO_2_ NPs synthesized from the leaf extract of *Ludwigia octovalvis* were found to show apoptotic activity against cervical carcinoma cell line (HeLa) [197]. The treatment of prepared TiO_2_ NPs resulted in potential cytotoxicity in cancer cells by increasing ROS generation and decreasing cell viability.

### 9.6. Plant-Based Iron Oxide NPs

Iron oxide nanoparticles (FeONPs) can be employed for a variety of medical applications, including MRI contrast enhancement [198], targeted drug delivery [199], cell separation and hyperthermia [200], detoxification of biological fluids [201], tissue healing, and cancer treatment [200]. The therapeutic potential of FeONPs for cancer treatment has been extensively studied. The beneficial properties of FeONPs make them an ideal tool for cancer therapy. These properties include: the stability of FeONPs, enhanced magnetic signal strength, low toxicity, eco-friendly, and their surface modification abilities [202,203,204]. There has been increased interest in creating clean, straightforward, low-cost, and environmentally friendly processes for synthesizing FeONPs using plant extracts for cancer therapies. Green synthesis of FeONPs using the leaf extract of *Solanum lycopersicum* has been found to show anticancer properties by inhibiting cell growth and inducing apoptosis in the human lung cancer A549 cell line [205]. In another study, the green synthesis of FeONPs synthesized from *Psoralea corylifolia* was found to inhibit the growth of cancer cells and induce apoptosis in renal carcinoma cells in a dose-dependent manner [202]. Green synthesis of magnetic FeONPs using the leaf extract of *Albizia adianthifolia* has been found to cause a cytotoxic effect and cell death via apoptosis in human breast (AMJ-13) and (MCF-7) cancer cells [206]. The treatment FeONPs of *Albizia adianthifolia* on cancer cells leads to DNA fragmentation, ROS production, loss of MMP, and membrane disintegration, thereby confirming apoptosis in AMJ-13 and MCF-7 cell lines. The synthesis of FeONPs (ferric oxide NPs) using the leaf extract of *Rhus punjabensis* has been investigated for their anticancer properties against HL-60 leukemic and DU-145 prostate cancer cell lines [207]. It was found that FeONPs of *Rhus punjabensis* induce a cytotoxic effect against cancer cells with an effective dose (ED_50_) of 11.9 and 12.79 μg/mL, respectively. The apoptotic potential of synthesized FeONPs was confirmed by inhibiting NF-κB inhibitory activity. In another study, the leaf and stem extract of *Artemisia absinthium* was used as a reducing agent for the synthesis of FeONPs, and its anticancer properties were evaluated [208]. It was found that FeONPs of *Artemisia absinthium* were effective for inducing cytotoxicity against melanoma A375 cells. The treatment of green synthesized FeONPs (500 µg/mL for a period of 72 h) resulted in a high release of lactate dehydrogenase (LDH) and apoptosis in A375 cells.

### 9.7. Plant-Based Quantum Dots

Quantum dots (QDs), commonly referred to as semiconductor nanocrystals, have been widely used in biology and medicine [209,210]. QDs may be used in various industrial and biomedical fields due to their distinctive electrical and optical properties [211,212,213]. At the cellular level, QDs can induce ROS generation and cell apoptosis, damage DNA, disrupt intracellular calcium signaling channels, suppress DNA repair, and induce cell death [214,215,216,217]. The green synthesis of QDs for anticancer activities has been the center of interest for many researchers across the globe. The green synthesis of cadmium sulfide (CdS) using tea leaf extract has been found to exhibit cytotoxicity and cell death via mechanism apoptosis in A549 cancer cells [218]. The green synthesized CdS QDs were found to inhibit the growth of cancer cells by arresting the growth of A549 cells at the S-phase of their cell cycle. Hydrothermal green synthesis of carbon quantum dots (CQDs) from walnut oil has been found to exhibit a cytotoxic effect on PC3, MCF-7, and HT-29 human carcinoma cell lines [219]. Upregulation of caspase-3 was observed in cancer cells, thereby confirming the induction of apoptosis by green synthesized CQDs on cancer cells. In another study, the roots of *Rhaphanus sativus*, used as reducing and stabilizing agents in the biosynthesis of CdS QDs, have been found to induce a cytotoxic and apoptotic effect on AGS gastric cancer and MCF-7 breast cancer cells [220]. CQDs synthesized from ginger extract have been found to suppress growth and induce apoptosis in human hepatocellular carcinoma cells (HepG2) [221].

**Table 3 biomolecules-13-00194-t003:** Plant-based nanoparticles and their cytotoxicity against human cell lines.

Type of NPs	Plant Name	Plant Part Used	Size and Shape of NPs	Cell Line	IC_50_	Ref.
AgNPs	*Litchi chinensis*	Leaf	41–55 nm (spherical)	MCF-7	40.9 μg/mL	[222]
AgNPs	*Putranjiva roxburgi*	Seed	8 nm (spherical)	MDA-MB-231, HCT-116, and PANC-1	260, 540, 0.25 μg/mL	[223]
AgNPs	*Alternanthera sessilis*	Leaf	30–50 nm (spherical)	PC3	6.8 μg/mL	[224]
AgNPs	*Euprenolepis procera*	Leaf	60 nm (spherical)	MCF-7	9.63 μM	[225]
AgNPs	*Solanum trilobatum*	Fruit	41.90 nm (spherical, polygonal)	MCF-7	30 μg/mL	[226]
AgNPs	*Zingiber officinale*	Leaf	18.93 nm (spherical)	AsPC-1, PANC-1, and MIA PaCa-2	295, 312, 220 µg/mL	[227]
AgNPs	*Punica granatum*	Leaf	46.1 nm	HeLa	100 μg/mL	[228]
AgNPs	*Ganoderma neo-japonicum*	Whole	5–8 nm (spherical)	MDA-MB-231	6.0 μg/mL	[229]
AgNPs	*Derris trifoliate*	Seed	16.92 nm (spherical)	A549	86.23 μg/mL	[230]
AgNPs	*Detarium microcarpum*	Leaf	81 nm (cubic)	HeLA, PANC-1	31.5, 84 μg/mL	[231]
AuNPs	*Gelidium pusillum*.	Whole	12 ± 4.2 nm (spherical)	MDA-MB-231	43.09 ± 1.6 μg/mL	[232]
AuPs	*Artocarpus hirsutus*	Leaf	5–40 nm (spherical)	HeLa, RKO, and A549	210, 210, 205 μg/mL	[233]
AuNPs conjugated with activated folic acid (FA, receptor) and chlorambucil	*Artocarpus hirsutus*	Leaf	5–40 nm (spherical)	HeLa, RKO, and A549	130, 132, 133 μg/mL	[233]
AuNPs	*Musa paradisiaca*	Peel	50 nm (triangular)	A549	58 μg/mL	[234]
AuNPs	*Ferula persica*	Gum	37.05 nm (spherical)	CT26	2.4 μg/mL	[235]
AuNPs	*Nigella sativa*	Essential oil from seed	15.6–28.4 nm (spherical)	A549	28.37 μg/mL	[236]
AuNPs	*Cajanus cajan*	Seed coat	9–41 nm (spherical)	HepG2	6 μg/mL	[237]
AuNPs	*Rhus chinensis*	Plant gall	20–40 nm (oval and spherical)	Hep3B	150 μg/mL	[238]
AuNPs	*Commelina nudiflora*	NA	24–150 nm (spherical, triangular)	HCT116	200 μg/mL	[239]
AuNPs	*Butea monosperma*	Bark	35 nm (spherical)	MCF7	0.024 μg/mL	[240]
CuNPs	*Olea europaea*	NA	20–50 nm (spherical)	SKOV-3	2.27 μg/mL	[186]
CuNPs	*Ficus religiosa*	Leaf	577 nm (spherical)	A549	200 μg/mL	[187]
CuNPs	*Olea europaea*		20–50 nm (spherical)	AMJ-13	1.47 μg/mL	[186]
CuNPs	*Azadirachta indica*, *Hibiscus rosa-sinensi, Murraya* *koenigii*, *Moringa* *oleifera*, and *Tamarindus indica*	Leaf	12 nm (spherical)	HeLa	26.3, 21.63, 23.22, 30.08, 20.32, μg/mL	[241]
		Leaf	12 nm (spherical)	MCF-7	25.55, 22.45, 25.32, 26.1, 29.1 μg/mL	[241]
		Leaf	12 nm (spherical)	A549	26.03, 20.15, 25.05, 34.3, 18.11 μg/mL	[241]
		Leaf	12 nm (spherical)	Hep-2	28.59, 22.59, 25.59, 29.58, 21.66 μg/mL	[241]
ZnO NPs	*Geranium wallichianum*	Leaf	18 nm (hexagonal)	HepG2	39.26 μg/mL	[242]
ZnO NPs	*Raphanus sativus*	Leaf	209 nm (spherical and hexagonal)	A549	40 μg/mL	[243]
ZnO NPs	*Rubia tinctorum*	Leaf	40 nm (spherical)	MCF-7	40 μg/mL	[244]
ZnO NPs	*Pongamia pinnata*	Seed	30.2 nm (face centered, cubic)	MCF-7	32.8 μg/mL	[245]
ZnO NPs	*Withania somnifera*	Root	32 nm (hexagonal wurtzite)	MCF-7	6.84 μg/mL	[246]
ZnO NPs	*Abutilon indicum*	Leaf	50–500 nm (spherical)	Hela	45.82 μg/mL	[247]
Fe_2_O_3_ NPs	*Albizia adianthifolia*	Leaf	32–100 nm (spherical)	AMJ-13, MCF-7	1.8, 7.7 μg/mL	[206]
Fe_2_O_3_ NPs	*Couroupita guianensis*	Fruit	17 nm (spherical)	HepG2	44.51 μg/mL	[248]
Fe_2_O_3_ NPs	*Sargassum muticum*	Whole	18 nm (cubic)	Jurkat, MCF-7, HeLa, and HepG2	6.4, 18.5, 12.5, 23.83 μg/mL	[249]
PtNPs	*Dioscorea bulbifera*	Tuber	2–5 nm (spherical)	HeLa	10 μg/mL	[250]
Palladium (Pd) NPs	*Dioscorea bulbifera*	Tuber	10–20 nm (spherical and blunt ended)	HeLa	10 μg/mL	[250]
Pt-Pd	*Dioscorea bulbifera*	Tuber	20–25 nm (irregular)	HeLa	10 μg/mL	[250]
Tin oxide (SnO_2_) NPs	*Annona squamosa*	Peel	2.5 nm (spherical)	HepG2	1–500 μg/mL	[251]

## 10. Conclusions

Targeting programmed cell death in rapidly dividing cancer cells is a potent strategy for combating cancer. Several PCs directly or indirectly target the apoptotic pathways and induce cell death through the activation of pro-apoptotic proteins or de-activation of anti-apoptotic proteins. This has already led to advanced research in the field of cancer biology. Even though the mechanism of action for these metabolites is exceedingly complex and displays certain on- or off-target activities, it is imperative to advance the plant cancer research beyond the existing utility. The availability of ethnopharmacological data offers an ideal advantage in plant-based drug discovery. To fully utilize PCs, a novel integrated drug discovery approach is required. This combines natural product chemistry with broad interdisciplinary forces from medicinal chemistry, pharmacology, biochemistry, molecular and cellular biology, and ethnopharmacology. Additionally, the development of artificial intelligence systems [252], genome editing technology [253,254], and computational methodologies will make it easier to find novel PCs for anti-cancer evaluation. The advancement in green synthesis for developing several plant-based NPs with applications in cancer therapy is also gaining a lot of attention. Plant-based NPs are not only effective at targeting the PCD pathway, but they also have higher rate of target specificity. These agents can be used in synergy with the existing cytotoxic agents, radiotherapy, and/or immune-therapeutic agents. The benefits and adaptability of carrier-free nanodrug systems for the treatment of cancer are gradually gaining importance. To increase drug delivery effectiveness, exert drug efficacy, and lessen toxic effects and side effects, NPs based on a phytochemical self-assembly strategy use bioactive natural drugs as the carrier without introducing other materials. Self-assembled NPs of various PC molecules can work to treat a wide range of cancer cells. Targeting apoptosis continues to be an efficient oncological strategy, and it will continue to advance in the future as well.

## Figures and Tables

**Figure 1 biomolecules-13-00194-f001:**
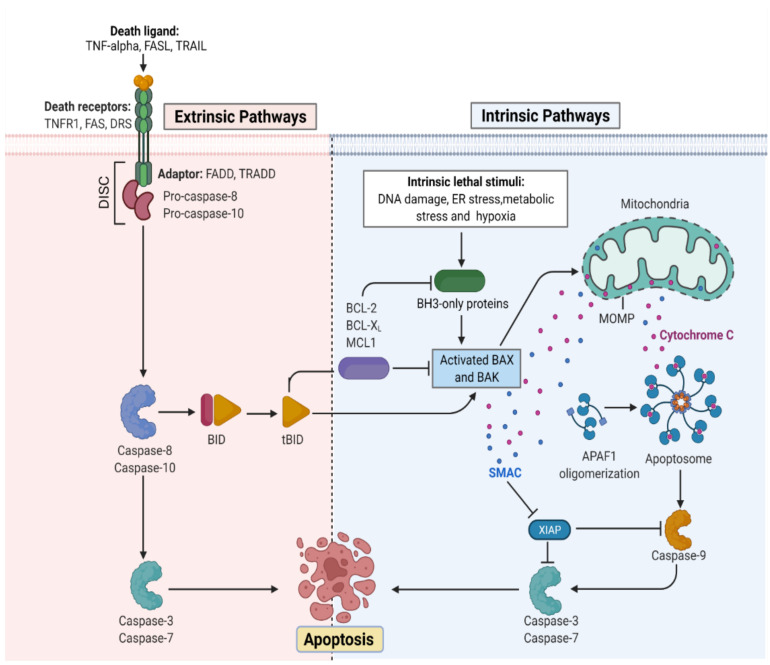
Representation of the molecular activation mechanism of intrinsic and extrinsic pathways of programmed cell death. Both intrinsic and extrinsic pathways depend on specific signals to initiate an energy-dependent cascade of molecular events, thereby activating initiator caspases which leads to the activation of executioner caspase-3.TNF (Tumor necrosis factor), FASL (FS-7-associated surface antigen ligand), TRAIL (Targeting TNF-related apoptosis-inducing ligand), TNFR1 (Tumor necrosis factor receptor 1), DRS (Death receptors), DISC (death-inducing signaling complex), BID (BH3 interacting-domain), MCL (myeloid leukemia cell differentiation protein), ER (Endoplasmic reticulum), BAX (Bcl-2-associated X protein), BAK (B-cell lymphoma 2 killer), SMAC (Second mitochondria-derived activator of caspase), XIAP (X-linked inhibitor of apoptosis protein), MOMP (mitochondrial outer membrane permeabilization), and APAF (Apoptotic protease activating factor). (Adapted from “Apoptosis Extrinsic and Intrinsic Pathways” by BioRender.com. (https://app.biorender.com/biorender-templates by BioRender.com, access on 30 November 2022).

**Figure 2 biomolecules-13-00194-f002:**
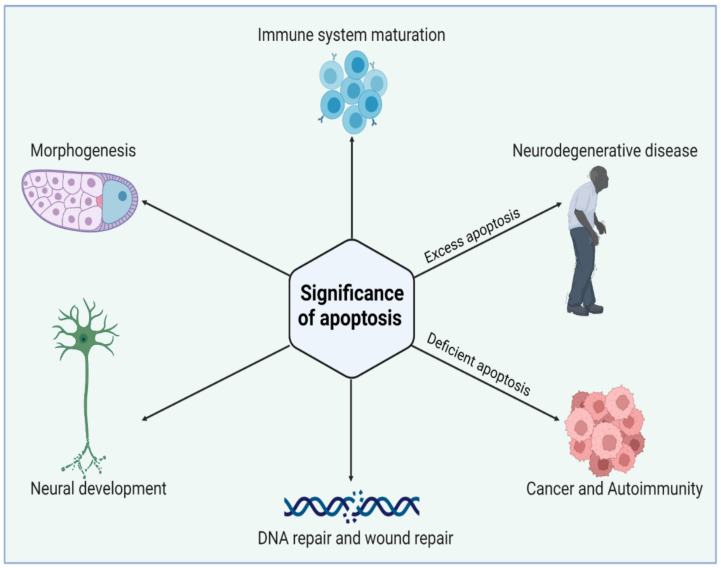
Biological significance of apoptosis in maintaining tissue homeostasis during cell life. Apoptosis has an essential role in the elimination of unwanted cells during early development. Excess or deficient apoptosis leads to several diseased conditions. Figure is created with BioRender.com (https://app.biorender.com, access on 22 November 2022).

**Figure 3 biomolecules-13-00194-f003:**
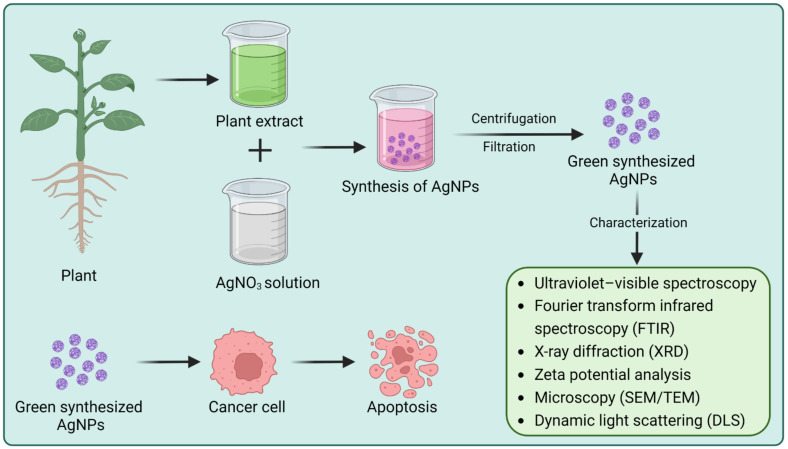
Scheme for green synthesis of silver NPs and their anticancer property. FTIR (Fourier transform infrared, SEM (scanning electron microscope), and TEM (transmission electron microscopy) Figure is created with BioRender.com (https://app.biorender.com, access on 22 November 2022).

**Table 1 biomolecules-13-00194-t001:** Role of PCs as anti-cancer agents through direct or indirect apoptotic mode of action.

Phytochemicals (Pcs) or Derivatives	Source	Cancer Type	Mode of Action	Ref.
Ursolic acid 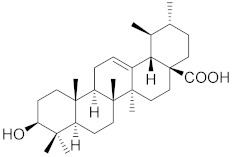	*Oldenlandia diffusa*	Lung cancer	Mitotic kinase activity inhibition	[71]
	*Punica granatum*	Breast cancer	Anti-cancer property via Akt/mTOR signaling	[72]
Apigenin 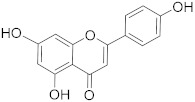	*Sorghum bicolor*	Colon cancer	Inhibiting ATP-binding cassette (ABC) transporter expression	[73]
Quercetin 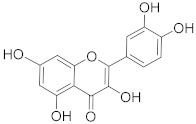	*Vigna unguiculat*	Colon cancer	Inhibiting multidrug resistance gene 1 (MDR1) expression	[74]
Kaempferol 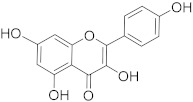	*Alangium salvifolium*	Ascitic lymphoma	Inhibition of dihydrofolate inhibition, thereby damaging DNA of a cancerous cell	[75]
Lupeol 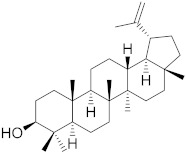	*Dillenia indica*	Leukemia	Induces PCD	[76]
Aurantio-obtusin 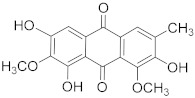	*Cassia tora*	Cervical cancer	Inhibits proliferation and induces apoptosis	[77]
Luteolin 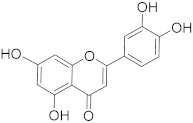	*Eclipta alba*	Glioma	Down regulating NF-κBleads to DNA damage and ultimately apoptosis.	[78]
Paniculatine 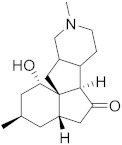	*Celastrus paniculatus*	Breast cancer	Growth inhibition followed by apoptosis mediated by p53 and mitogen activated protein (MAP) kinases	[79]
Linalool 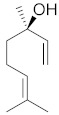	*Cinnamomum cassia*	Cervical carcinoma	Reduced expression of Her-2 oncoprotein	[80]
Carthamin 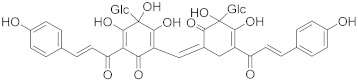	*Carthamus tinctorius*	Colon cancer	Caspase-3, -7, and -9 upregulation and Bcl-2 downregulation	[81]
Resveratrol 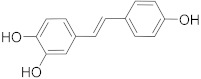	*Vitis vinifera*	Colorectal cancer	Regulation of p53 and inhibition of IGF-IR pathway	[82]
Fisetin 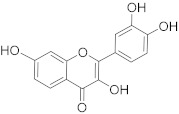	*Fragaria ananassa*	Hepatic cancer	Modulated mitogen activated protein kinases (MAPK) and CDK5 signaling	[83]
Withaferin A 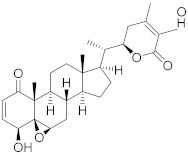	*Withania somnifer*	Colon cancer	Inhibition of (signal transducer and activator of transcription 3) STAT3 pathway	[84]
Ferulic acid 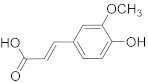	*Ferula asafetida*	Mammary carcinogenesis	Reduction in cytochrome p450 levels	[85]
Safranal 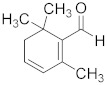	*Crocus sativus*	Cervical cancer	Inhibition of malignant cell line proliferation and induces apoptosis	[86]
Limonene 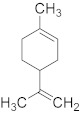	*Citrus sinensis*	Breast cancer	Induction of apoptosis through upregulation of pro-apoptotic factors and down-regulation of anti-apoptotic factors	[87]
Gingerol 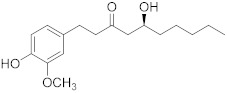	*Zingiber officinale*	Colon cancer	G2/M cell cycle arrest, activation of PI3K/Akt pathway, and apoptosis induction	[88]
Malic acid 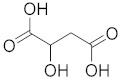	*Actinidia deliciosa*	Brain tumor	Activates pro-apoptotic genes	[89]
Podophyllotoxin 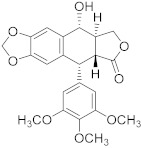	*Podophyllum* spp.	Testicular and breast cancer	Cell division blockage at metaphase of mitosis	[90]
Berberine 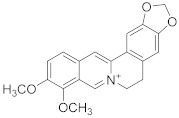	*Berberis vulgaris*	Colon cancer	Cell cycle arrest and synergistic effect with drugs	[91]
Combretastatin 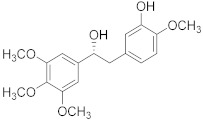	*Combretum caffrum*	Leukemia	Destabilization of microtubules	[92]
Ingenol mebutate 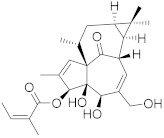	*Euphorbia peplus*	Prostate cancer	Suppression of P13/Akt pathway	[93]
Genipin 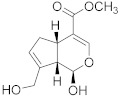	*Gardenia jasminoides*	Medullobalstome	Increase in Bax level	[94]
Crocin	*Crocus sativus*	Prostate cancer	G2/M Cell cycle arrest	[95]
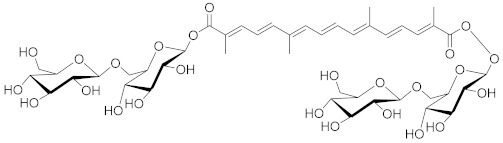				
Bilobetin 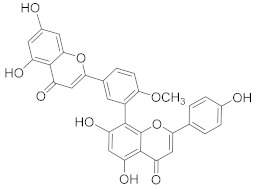	*Ginkgo biloba*	Leukemia	Cell cycle arrest at G2/M	[96]
Amentoflavone 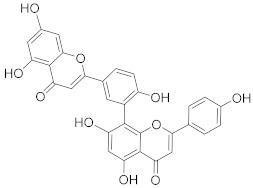	*Selaginella tamariscina*	Ovarian cancer	Inducing DNA damage by interfering in microtubule dynamics	[97]
Brusatol 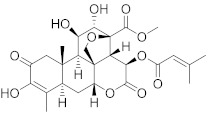	*Brucea javanica*	Lung cancer	Stimulation of ROS; G0-G1 cell cycle arrest and decrease in glutathione levels	[98]
Brucea javanica oil	*Brucea javanica*	Hepatocellular carcinoma	Upregulation of miRNA-29b and p53 expression levels	[99]
Octahydrocurcumin 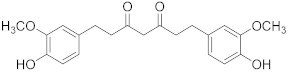	*Curcuma longa*	Hepatocellular carcinoma	Downregulation of murine double minute 2 (MDM2) expression levels	[100]

**Table 2 biomolecules-13-00194-t002:** Combinatorial effect of PCs and conventional anti-cancer drugs.

Phytochemicals (PCs)	Chemotherapeutic Drug	Cancer Cell Target	Combinatorial Effect	Ref.
Emodin 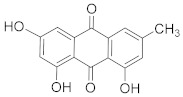	Cisplatin	Non-small cell lung cancer	Enhances cisplatin sensitivity through p-glycoprotein downregulation	[103]
Caffeic acid 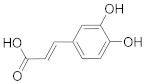	Paclitaxel	Lung cancer	Increase in the activities of caspase-3 and -9; activation of Bax and Bid	[104]
Diosmetin 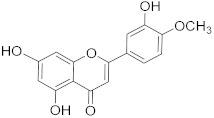	Paclitaxel	Lung cancer	Increases therapeutic efficacy inducing ROS production, thereby disrupting PI3K/Akt pathway.	[105]
Curcumin 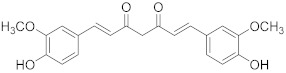	Docetaxel and gemcitabine	Pancreatic cancer	Triggers apoptosis through caspase/PARP signaling pathway	[106]
Baicalein 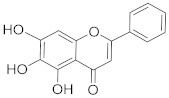	Gemcitabine and docetaxel	Pancreatic cancer	Cell cycle arrest in S phase	[107]
Genistein 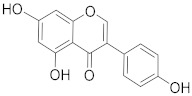	Gemcitabine	Pancreatic cancer	Inactivation of NF-kappaB signaling pathway	[108]
Berberine 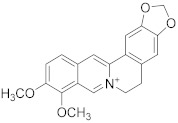	Cisplatin	Breast cancer	Induction of DNA breaks and upregulation of Bcl-2	[109]
		Ovarian cancer	G0/G1 cell cycle arrest	[110]
		Osteosarcoma	Inhibition of MAPK pathway	[111]
Oridonin 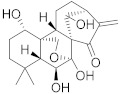	Cisplatin	Ovarian cancer	Reverses cisplatin drug resistance and thereby induces apoptosis	[112]

## Data Availability

No new data were created or analyzed in this study. Data sharing is not applicable to this article.

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
