# Peer review of "Targeting Apoptotic Pathway of Cancer Cells with Phytochemicals and Plant-Based Nanomaterials"

_biomolecules, 2023, doi:10.3390/biom13020194_

Round 1

Reviewer 1 Report

This review provides detailed insight into the fundamental molecular pathways of programmed cell death and highlights the role of PCs along with the existing drugs and plant-based NPs in treating cancer by targeting its programmed cell death (PCD) network.

The review is well-written and discussed the topic in detail. However, I recommend minor revisions as follows;

- The abstract should be more concise.

- The conclusions section should clearly highlight the main conclusions and mention the recommended future perspectives.

- The reference section has some old references that are not required indeed. I suggest that the authors should furnish or delete them.

Reviewer 2 Report

The authors propose “to compile the role of PCs in inducing apoptosis individually, synergistically, and in association with designed NPs thereby, opening new avenues in cancer pathology concerning cancer treatment.”

Title: 

The title of the manuscript is concise, specific, and relevant.

Abstract:

The abstract should be a total of about 200 words maximum and it has 208 words. I suggest authors adhere to the journal's instructions.

Introduction

The introductory section is well organised by first describing cancer and the importance of apoptosis in the aetiology and progression of the disease. Second, the apoptotic pathways and the ability of cancer cells to disrupt these pathways are described in general terms. The rationale for the use of new alternative therapies such as plant-derived products is also highlighted. Third, the importance of the use of plant-derived products in cancer treatment in the last decade is highlighted, which justifies the objective of this review “to compile the role of PCs in inducing apoptosis individually, synergistically, and in association with designed NPs thereby, opening new avenues in cancer pathology concerning cancer treatment.”

Lines 58-59: I recommend that the authors review the rules of punctuation and consider changing the full stop followed by a full stop to a full stop to reduce the length of the paragraph for better comprehension.

Lines 71-72: I recommend that the authors review the rules of punctuation and consider changing the full stop followed by a full stop to a full stop to reduce the length of the paragraph for better comprehension.

Development and discussion

Line 87: I recommend that the authors consider editing the title or content of this section, as it sets out to describe the molecular perspective and 60% of the information refers to the cellular perspective. Perhaps it could read “Molecular and cellular perspective”

Lines 97-98: The authors claim that apoptosis is an irreversible process, but I think it would be more appropriate to refer to a moment in the imbalance between death and survival known as the "point of no return" on the way to death, characterised by the release from the intermembrane space into the cytosol of proteins with key pro-apoptotic functions such as: cytochrome c, apoptosis-inducing factor, endonuclease G, etc.

Lines 132-165: This activation pathway is partially described in section “3.1. Mitochondria pathway of cellular apoptosis”. I recommend the authors not only to mention MOMP, but also MIMP, which are related but molecularly different processes within the mitochondrial apoptosis pathway.

Lines 193-305: I suggest that the authors consider revisiting the usefulness of all this information for the main purpose of this review “to compile the role of PCs in inducing apoptosis individually, synergistically, and in association with designed NPs thereby, opening new avenues in cancer pathology concerning cancer treatment.” I believe that the reader, when encountering this review, will be motivated to aim and not read more about apoptosis. In addition, there are a variety of excellent reviews that describe in depth the topics described in these paragraphs. The authors could introduce these topics and invite authors to read them.

Lines 306-402: The authors invite the reader to learn about the apoptotic pathway targets of some Phytochemicals and Plant-based Nanomaterials", but it seems that not all these compounds are specifically targeted by an apoptotic pathway or molecule involved in this pathway.  It may be appropriate to specify the target apoptotic pathway of each molecule or consider editing the title.

Lines 492-493: The authors claim that “The biosynthesized NPs inhibited lipid peroxidase-induced ROS production”, but I think the process is the reverse, ROS generates lipid peroxidation. I recommend the authors to review this information.

In several paragraphs of the manuscript the authors mention the effects of plant-derived products on ROS production, but we should not forget that the mitochondrion is not the only source of ROS production in the cell, but also at the plasma membrane level, it would therefore be interesting to indicate the origin of ROS production if this information is available in the literature cited, e.g. lines 495-496, lines 523-524, line 558, line 576, line 585, line 589, line 600, line 623, line 638, etc. It seems that some products are rather inducers of oxidative stress, which in the second instance activate apoptotic cell death, but do not target the previously mentioned apoptotic pathways.

Reviewer 3 Report

Dear authors,

you have prepared the review with very interesting and actual topic. The review is written very well but I have some recommendation to improve it:

1. Introduction:

You have cited realativelly old cancer statistic, reference no 1. I recommend to find and cite some statistics from last few years.

2. Introduction:

Last part of the intorduction contains relativelly low number of references. For example this sentence "There have been several articles published that have highlighted the role of PCs as  anti-cancer agents." is withou any reference. I can recommend for example this article as a reference for this part:

Fatima et al. Phytochemicals from Indian Ethnomedicines: Promising Prospects for the Management of Oxidative Stress and Cancer. Antioxidants. 2021.

3. Chapter 2: Apoptosis: Molecular perspective

Reference 14 is old. I recommend to use some recent reference.

4. term "nanoparticles": If you use abbreviation NPs, use it in all manuscript.

Generally, it is very good review.

I recommend publication after minor revisions following suggested changes.
